# Rotation of Liquid Metal Droplets Solely Driven by the Action of Magnetic Fields

**Jian Shu** [1], **Shi-Yang Tang** [2,\*], **Sizepeng Zhao** [1], **Zhihua Feng** [1], **Haoyao Chen** [3], **Xiangpeng Li** [4,\*], **Weihua Li** [2] and **Shiwu Zhang** [1,\*]

1   CAS Key Laboratory of Mechanical Behavior and Design of Materials, Department of Precision Machinery and Precision Instrumentation, University of Science and Technology of China, Hefei 230026, China; jianshu@mail.ustc.edu.cn (J.S.); zszp894345865@mail.ustc.edu.cn (S.Z.); fff@ustc.edu.cn (Z.F.)
2   School of Mechanical, Materials, Mechatronic and Biomedical Engineering, University of Wollongong, Wollongong, NSW 2522, Australia; weihuali@uow.edu.au
3   School of Mechanical Engineering and Automation, State Key Laboratory of Robotics and System, Harbin Institute of Technology Shenzhen Graduate School, Shenzhen 518055, China; hychen5@hit.edu.cn
4   College of Mechanical and Electrical Engineering, Soochow University, Suzhou 215006, China
\*   Correspondence: swzhang@ustc.edu.cn (S.Z.); shiyang@uow.edu.au (S.-Y.T.); licool@suda.edu.cn (X.L.); Tel.: +86-551-63600249 (S.Z.)

**Abstract:** The self-rotation of liquid metal droplets (LMDs) has garnered potential for numerous applications, such as chip cooling, fluid mixture, and robotics. However, the controllable self-rotation of LMDs utilizing magnetic fields is still underexplored. Here, we report a novel method to induce self-rotation of LMDs solely utilizing a rotating magnetic field. This is achieved by rotating a pair of permanent magnets around a LMD located at the magnetic field center. The LMD experiences Lorenz force generated by the relative motion between the droplet and the permanent magnets and can be rotated. Remarkably, unlike the actuation induced by electrochemistry, the rotational motion of the droplet induced by magnetic fields avoids the generation of gas bubbles and behaves smoothly and steadily. We investigate the main parameters that affect the self-rotational behaviors of LMDs and validate the theory of this approach. We further demonstrate the ability of accelerating cooling and a mixer enabled by the self-rotation of a LMD. We believe that the presented technique can be conveniently adapted by other systems after necessary modifications and enables new progress in microfluidics, microelectromechanical (MEMS) applications, and micro robotics.

**Keywords:** liquid metal; self-rotation; mixer; Lorentz force; EGaIn

## 1. Introduction

"Liquid metal" such as gallium and its several alloys exhibit a liquid state at room temperature [1–3]. Droplets of such liquid metal have shown to be platforms for applications such as stretchable and reconfigurable electronics [3–6], microfluidics actuators [2,7,8], as well as forming three-dimensional structures [9–11]. Liquid metal droplets (LMDs) have demonstrated many unique properties, for instance, large surface tension, favorable thermal and electrical conductivity, strong stability, and extremely excellent biosafety compared with mercury [12,13]. Over the past few years, benefitting from the flexibility and operability of liquid metal surfaces, investigations of formation, actuation, and application of their droplets have gained momentum [2,14–23]. As has been recently reported, LMDs can act as self-fueled motors, paving the way without human intervention [14,23]; also, a LMD can be used to drive a wheeled robot by changing its center of gravity under electrical fields [22]. Moreover, a potential gradient was applied to explore inducing the Marangoni flow along the surface of

LMDs and making microactuators in microfluidics [2,24–27]. Further studies have demonstrated that LMDs coating modest nanoparticles can be propelled by bubbles generated through photochemical reactions [28–30].

Chaotic advection is the key mechanism for enabling applications such as heat transfer, fluid mixing, and chemical reaction improvement [24,31–35], in particular, in some areas including microfluidic systems, chemical and biological transport, etc., smooth and steady methods for inducing chaotic advection are needed, and inexpensive and simple systems are urgently required [36–39]. Self-rotation of LMDs with negligible solubility in most solvents may be a promising candidate for providing a solution [7]. Nevertheless, to the best knowledge of the authors, there is still a lack of investigations focused on the self-rotational motion of LMDs.

Self-rotation of LMDs has the potential to be widely used in fluid cooling and mixing. Therefore, we have been motivated to explore novel methods for inducing self-rotation of LMDs that are smooth, simple, steady, and, especially, that do not introduce undesired violent chemical reactions. According to our previous report, we introduced a simple and violent chemical reaction-free method in which we utilized Lorenz force induced by magnetic fields to control the locomotion of LMDs [38]. Inspired by this, we report here the self-rotation of EGaIn droplets (consisting of 75% gallium and 25% indium) which are solely driven by magnetic fields without mixing or coating ferromagnetic particles. The relative motion of the magnetic fields and LMDs generates an eddy current in the droplet and further induces the Lorenz force to actuate the self-rotation of the droplet. The theory behind the method was developed and the experiments conducted to validate this approach. Moreover, we demonstrate applications of accelerating cooling and mixing liquids based on the self-rotational LMDs.

## 2. Materials and Methods

*Materials and instrumentation:* EGaIn liquid metal and sodium hydroxide (NaOH) were purchased from Santech Materials Co. Ltd, China. An electrolyte solution of NaOH and hydrochloric acid (HCl) were introduced to remove the oxide layer on the surface of LMDs, where NaOH solution was prepared by dissolving solid sodium hydroxide particles in deionized (DI) water, and HCl solution was prepared by diluting concentrated hydrochloric acid with DI water. A small amount (~2 mg) of fine phosphors (Juen technology Co. Ltd, China) was sprinkled on the LMDs to create easily identifiable points for calculating the rotational speed of the LMDs. Blue and yellow dyes were obtained by diluting edible dye (1:5, SUGARMAN) and then dripped into a quartz tube using a syringe pump (LSP02-1B, LONGERPUMP). Self-rotation videos of LMDs were captured using a digital single lens reflex camera (Canon 5D Mark II) equipped with a macro lens (Sigma 105mm 1:2.8 DG Macro HSM). The sequential snapshots were extracted from these videos.

*Experimental Setup:* The experimental setup is illustrated in Figure 1a. A pair of permanent magnets were fixed to an aluminum frame, and the aluminum frame was connected to the output shaft of a DC motor (Leadshine 57HS09) whose speed and direction were controlled by a microcontroller unit (MCU, Arduino Carduino UNO R3). An EGaIn LMD was placed in a quartz tube (diameter: 8 mm, height: 15 mm) filled with NaOH or HCl solution, and the quartz tube was placed at the center point between the magnets.

*Mechanism*: The mechanism of self-rotation of LMDs is shown in Figure 1b, in which we hypothesize that by moving two permanent magnets around the droplet within an aqueous solution, an EGaIn droplet can be actuated to self-rotate after the introduction of eddy current. On account of the large surface tension of liquid metal, the EGaIn droplet (diameter < 5 mm) immersed in the aqueous solution can be considered as a sphere. In addition, to simplify the model, the EGaIn droplet is equivalent to several parallel coils located at a different longitude of the droplet. Before the external magnetic field rotates, the magnetic flux through the equivalent coils is almost a constant. When the external magnetic field starts to rotate, relative motion (in other words, phase shift) is formed between the magnetic field and the EGaIn droplet, which further induces the change of the magnetic

flux through the equivalent coils $\varphi$ and eddy current $I$ within the equivalent coils. The eddy current $I$ can be explicitly expressed as

$$I = \frac{d\Phi}{R_e dt} \tag{1}$$

where $\varphi$ is determined by the magnetic flux density **B** as well as the equivalent coils area **S**, which can be written as $\varphi = \mathbf{B \cdot S}$, and $R_e$ represents the equivalent resistance of the coils. Subsequently, an induced magnetic field is engendered by the eddy current to hinder the change of magnetic flux, according to the Lenz's law [40,41]. In other words, assuming that the external magnetic field rotates counterclockwise, the induced magnetic field will generate a clockwise torque to hinder the rotation of the magnetic field, no matter whether the magnetic flux through the coils increases or decreases. It is well known that forces are mutual actions of two bodies [38,42–46], that is, when the equivalent coils impede the external magnetic field, the external magnetic field also exerts a torque **M$_e$** of the equal magnitude and opposite direction on the EGaIn droplet. According to Ampere's law, the torque **M$_e$** can be expressed as

$$\mathbf{M_e} = 2 \int_0^{\pi R} \sin \frac{l}{R} I dl \times B \tag{2}$$

where $R$ is the radius of the equivalent coil, and **l** is the length of the equivalent coil, respectively [42–46]. The torque **M$_e$** enables the droplet to overcome the friction and commence to self-rotate.

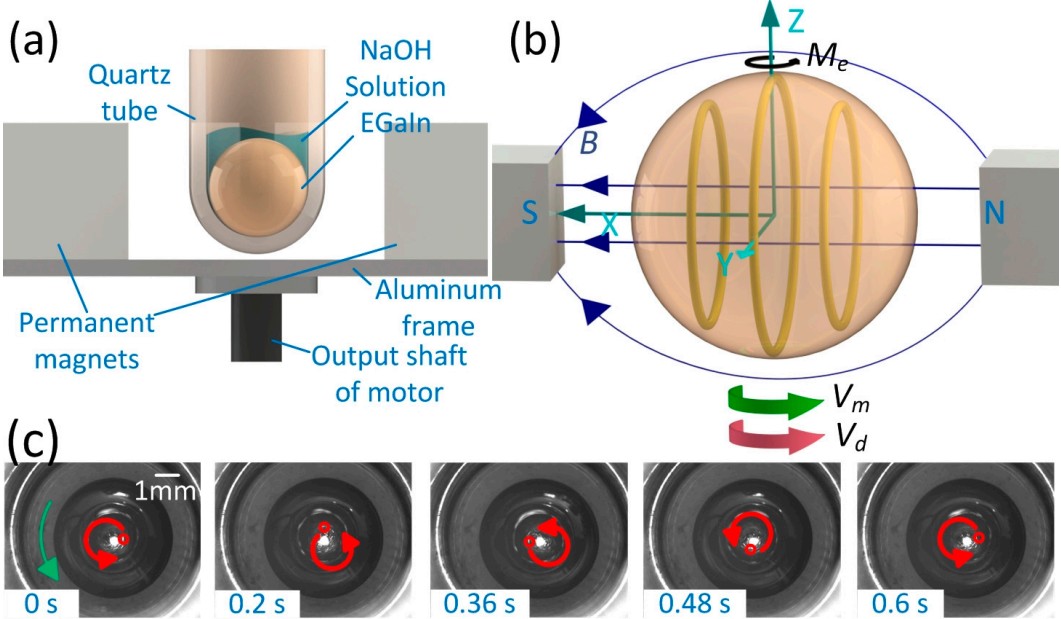

**Figure 1.** Self-rotation behavior of a liquid metal droplet (LMD). (**a**) Schematic of the self-rotation inducing setup. (**b**) Forces schematic of the equivalent coils in an EGaIn droplet; the magnetic field directions are indicated by the blue arrows, the magnetic field rotating direction (with a velocity of $V_m$) is indicated by the green arrow, the self-rotation direction of the EGaIn droplet (with a velocity of $V_d$) is indicated by the red arrow, and the forces experienced on the equivalent coils are indicated by the black arrows. (**c**) Continuous captures of the self-rotation of an EGaIn droplet (volume of 0.08 mL).

## 3. Results and Discussions

We established an actuating platform to examine the hypothesis and investigate the self-rotational behaviors of the droplet (Figure 1a). Figure 1c shows the self-rotation of an EGaIn droplet submerged in 0.5 mol/L NaOH solution induced by the magnetic field (also see Video S1); the measured magnetic flux density was ~1 kGs at the center point between the magnets, and we set the DC motor speed to 420 RPM. Then, the EGaIn droplet self-rotated following the same direction of rotation as the magnets (counterclockwise) with a speed of ~100 RPM, which aligns with our analysis given in Figure 1b.

On the basis of successfully demonstrating of magnetic field driven self-rotation of EGaIn droplet, we carried out a set of experiments to ascertain the performance of self-rotation. We found that motor speed, the LMDs size, as well as the NaOH solution concentration are the three main factors that affect the self-rotational speed. In Figure 2a, we can see that the self-rotational speed increases along with the increase of the DC motor rotational speed since larger rates of magnetic flux change $d\varphi$ can be induced by higher motor speeds. A larger rate of magnetic flux change further induces a larger eddy current which eventually converts into a greater $\mathbf{M_e}$ (see Equation 2). We also studied the influence of the magnetic flux density $\mathbf{B}$ on the self-rotational performance, as shown in Figure 2a. Obviously, a higher $\mathbf{B}$ can induce a larger torque and lead to a larger self-rotational speed.

Figure 2b shows that the self-rotational speed increases along with the increase of the size of the EGaIn droplets until the droplet volume reaches ~0.06 mL, and then the speed gradually decreases and eventually remains stable. This is probably due to that, in a larger droplet, the increasing $\mathbf{S}$ and decreasing $R_e$ enlarge the driving force. When the volume of the EGaIn droplets is larger than 0.06 mL, the friction between the droplet and the sidewall may give a negative effect on the rotational speed of the droplet which has been elucidated by repeating the experiment with a bigger tube, as discussed in Supporting Information S1. As the rotational speed decreases, the relative rotational speed of the droplet and the magnetic field increases (that is, $d\varphi$ increases), which then can cause the increase of $I$, the driven force, and the rotational speed. Next, increasing the rotational speed reduces the relative speed, and finally, the driving force and resistance are balanced, which is manifested in the fact that the rotation speed was basically stable during our experiments. Moreover, the depth of EGaIn droplet immersion in the NaOH solution (represented by the percentage of the height of the droplet immersed) also influences the speed of rotation (Figure 2b). For smaller droplets (<0.06 mL), the difference in rotational speed is not obvious for different immersion depths. However, for droplets larger than 0.06 mL, a larger immersion depth leads to a higher speed of rotation. This is probably due to the fact that despite the increase in viscous friction for larger immersion depths, the solid oxide layer formed on the EGaIn droplet surface can be efficiently removed by NaOH solution, thus a slip layer between the droplet and the container wall is formed. This slip layer can reduce the friction like a lubricant, and thus increase the rotational speed. Interestingly, we observed that when the droplet was 100% immersed, the droplet reached its maximum rotational speed when the volume was increased to 0.04 mL, which is smaller than that of other immersion depths (0.06, 0.06, and 0.08 mL). We believe this is due to the fact that the significant increase in rotational speed at 100% immersion flattens the droplets and, therefore, increases the friction between the droplets and the tube.

As shown in Figure 2c, NaOH concentration also affects the self-rotational speed of the LMDs. We found that using NaOH solution with a high concentration can lead to a faster rotational speed until the concentration of NaOH solution reaches 0.03 mol/L. When the concentration of NaOH solution exceeds 0.03 mol/L, the rotational speed of the droplets no longer increases with the increase of concentration and remains almost constant. No self-rotation was observed in our experiments when we reduced the NaOH solution concentration to zero. That might be due to the fact that without NaOH, the oxide layer cannot be removed, and the droplet becomes wrinkled [14,17], so the friction between the droplet, the tube, and the solution eventually increases. With the increase of the concentration of NaOH solution, the oxide layer was gradually removed, the friction gradually decreased, and the rotational speed increased. However, when the concentration reaches the threshold (0.03 mol/L), the oxide layer is almost completely removed, and the speed no longer increases with the concentration increase. Considering HCl solution can also be used to remove the oxide layer on the surface of EGaIn droplets [17,47], as shown in Figure 2d, we further studied the influence of HCl concentration on the rotational speed of EGaIn droplets. We observed the increase in rotational speed when increasing the concentration of HCl solution from 0 to 0.5 mol/L.

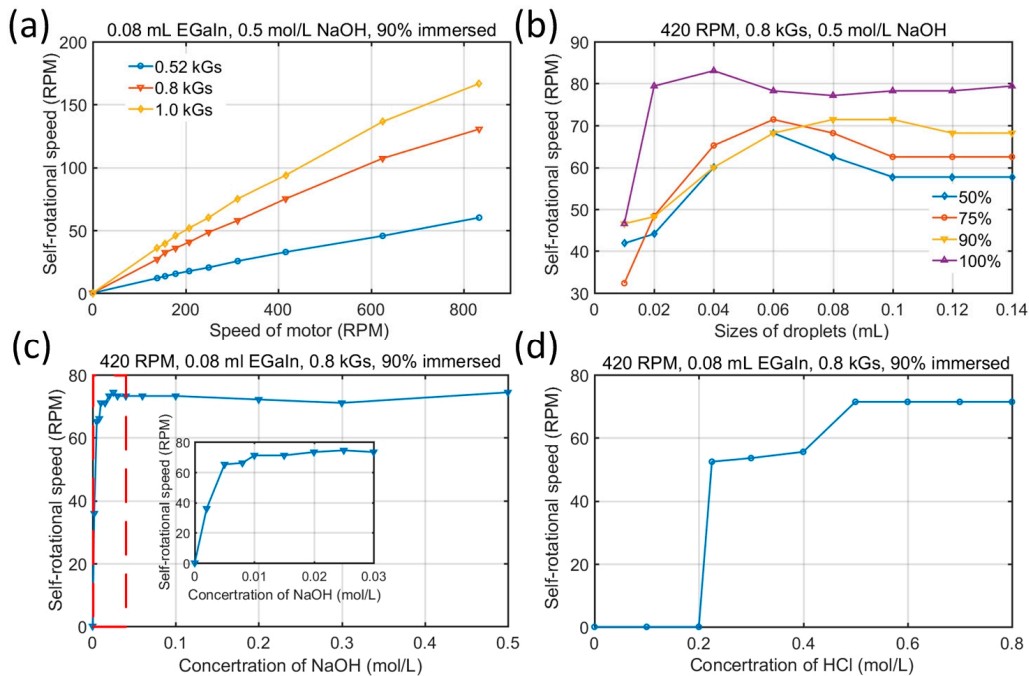

**Figure 2.** Change of the self-rotational performance under different operating parameters. (**a**) Self-rotational speed vs. speed of the motor plot; blue, red, and yellow curves indicate different magnetic field densities at the center of EGaIn droplets, respectively. (**b**) Self-rotational speed vs. sizes of droplets plot; blue, red, yellow, and purple curves indicate different depths of EGaIn droplet immersion in the NaOH solution, respectively. (**c**) Self-rotational speed vs. concentration of NaOH plot. (**d**) Self-rotational speed vs. concentration of HCl plot.

Use in cooling systems is an import potential application of liquid metal, and here we demonstrate the ability of liquid metal self-rotation to accelerate liquid cooling. As shown in Figure 3a, we heated the 0.5 mol/L NaOH solution and 0.08 mL EGaIn in a tube with a heat gun (DH-HG2-2000, Delixi Electric). When the temperature of the solution reached about 40 °C, a large number of bubbles were separated from the solution similar to boiling. We stopped heating and rotated the motor at 420 RPM and measured the temperature of the solution every 5 seconds. For comparison, the other tube was tested in the same way except that the permanent magnets in the device were removed, that is, the LMD did not self-rotate as the motor rotated. The temperature change is shown in Figure 3b; the group in which the EGaIn droplet self-rotated cooled significantly faster than the group in which the EGaIn droplet did not rotate. After about 520 s, the group of rotated EGaIn cooled to room temperature (~24.2 °C), and after another 130 s, the other group cooled to room temperature.

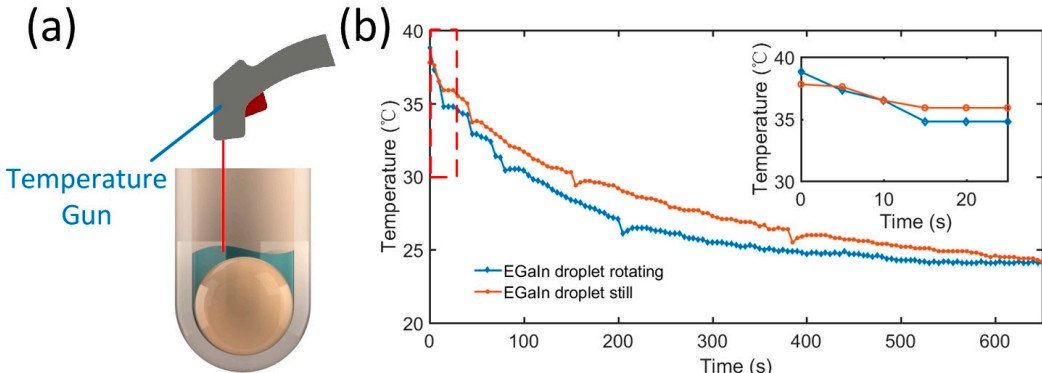

**Figure 3.** Cooling system based on a self-rotating LMD. (**a**) Schematic illustration of the cooling setup. (**b**) Temperature of solution vs. time plot.

Based on the fact that LMDs are negligibly soluble in most liquids, possess great surface tension, and keep almost unmixed with the solvent [7], LMDs have great potential for application as a mixer. Therefore, we further investigated its application as a mixer based on the self-rotation phenomenon of LMDs in a rotating magnetic field, as shown in Figure 4a (see also Video S2). We applied the self-rotation of EGaIn droplets to mix two droplets of the same volume, similar viscosity, and different colors. We added a drop of blue dye and a drop of yellow dye (volume of 0.05 mL, respectively) into the solution while the 0.08 mL EGaIn droplet was rotating in the NaOH (0.5 mol/L) solution, as shown in Figure 4b. The motor speed was 420 RPM, and the magnetic flux density at the center of the droplet was ~0.8 kGs. The self-rotational speed of EGaIn droplet after mixing was ~70 RPM which is almost the same as that when the EGaIn droplet is not acting as a mixer, as shown in Figure 2b. Figure 4c,d demonstrates that the two drops of dye can be mixed along with the rotating EGaIn droplet. This mixer is smooth, quick, and can be easily implemented into a MEMS platform.

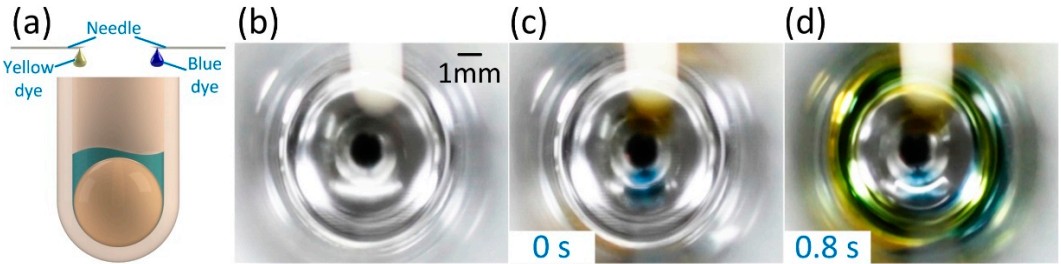

**Figure 4.** Mixer based on a self-rotating LMD. (**a**) Schematic illustration of the mixing setup. (**b–d**) Continuous captures of mixing two drops of dye.

## 4. Conclusions

We have demonstrated a novel method to manipulate the self-rotation of LMDs by solely utilizing magnetic fields which is smooth, simple, steady and, most importantly, does not produce a violent chemical reaction. The relative motion of the magnetic fields and LMDs generates an eddy current in the droplets and further induces the Lorentz force to actuate the self-rotation of the droplets. The motor speed, the LMD sizes, and the concentration of NaOH solution can be easily manipulated to regulate the self-rotational speed of droplet. Moreover, we demonstrated that such a technique can be used for the application of accelerating the cooling and mixing liquids. Nonetheless, bulky rotating magnets and a motor are still required in our current platform; however, we believe that introducing programmed electromagnetic fields can be helpful in resolving this problem in our future work. As such, utilizing magnetic fields to induce the self-rotation of LMDs could widely expand the application of LMDs to be used as MEMS devices.

**Supplementary Materials:** The following are available online at http://www.mdpi.com/2076-3417/9/7/1421/s1, Figure S1: Self-rotation of EGaIn droplets in different diameter tubes, Video S1: Self-rotation of liquid metal droplet, Video S2: Mixer based on self-rotation of liquid metal droplet.

**Author Contributions:** S.J., the first author, conceived the idea, performed the experiments, analyzed the data and wrote the paper. S.-Y.T. conceived the idea, analyzed the data and wrote the paper. S.Z. (Sizepeng Zhao) performed the experiments. Z.F. conceived the idea and analyzed the data. H.C. conceived the idea. W.L. analyzed the data and wrote the paper. X.L. conceived the idea, analyzed the data and wrote the paper. S.Z. (Shiwu Zhang) conceived the idea, analyzed the data and wrote the paper.

**Funding:** The authors acknowledge support from the National Natural Science Foundation of China (No. 51828503, U1713206, 61503270, 61873339), and the Shenzhen Science and Innovation Committee (Reference No. JCYJ20160427183958817). Dr. Shi-Yang Tang is the recipient of the Vice-Chancellor's Postdoctoral Research Fellowship funded by the University of Wollongong.

**Conflicts of Interest:** There are no conflicts of interest to declare.

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
