# Peer review of "Rotation of Liquid Metal Droplets Solely Driven by the Action of Magnetic Fields"

_applsci, doi:10.3390/app9071421_

Round 1
Reviewer 1 Report
The manuscript presents a novel technique for rotating LMD utilizing the internal eddy currents produced by a rotating external electric field. The advantage of the herein proposed method is the complete reliance of action at a distance, meaning that the LMD rotating system is chemically, mechanically and thermodynamically isolated from the drive system. This work could prove to be a vital step in the development of future technologies and the authors envision refinements on the current design which would prove compatible with current lab-on-a-chip and microfluidic technologies.
As a proof of concept the report is of importance and should be suitable in a journal such as Applied Sciences. There are however several issues which should be addressed prior to consideration for publication.
1. Materials:
Authors should outline here what the electrolyte solutions are (e.g. NaOH in DI).
2. Size dependency experiment, 117-126, Figure 2b:
The interpretation that the rotational friction against the sidewall is unsupported, repeating the experiment with different geometry test tubes would elucidate this.
3. Immersion dependency experiment, 126-136, figure 2b:
More evidence is required. If the Authors claim is true then the speed should maximize at or over 100% immersion as the ‘slip’ increases and until reducing with the increased viscous friction.
4. Elecrolyte concerntration expirement, 135-143, Figure 2 c and d
More data points are required.
I do not believe the author’s interpretation explains the phenomena fully. There are insufficient Data points to properly determine response of the system.
a. I agree that the oxidation prevention is the likely the key mechanism increasing speed however, the authors use vague imprecise terminology.
i. I would assume the response to appear to be that of a step response (V = Vmax(1-e^(-x/c)), where x is the concentration and c a constant based on the friction co-efficient of the oxide and the equilibrium oxidation in the solution.
ii. It also seems that HCl has a minimum concentration.
5. 138-149: I believe you mean to say ‘NaOH water was substituted for DI water’ the way it is written currently is confusing to the reader. Although in context simply stating that NaOH concentration is reduced to zero is sufficient for the reader if the solution is previously defined (see earlier comment).
6. Mixing experiment, Figure 4, 151-157: were different volumes, viscosities of mixing liquid?
a. How would this affect the speed of rotation?
7. As the authors mention cooling as one potential application the thermal respose of the system should be tested.
Author Response
Response to Reviewer 1 Comments
1. Materials:
Authors should outline here what the electrolyte solutions are (e.g. NaOH in DI).
Response 1:
Thank you for your suggestion and the following descriptions have been added to the main manuscript to make clear the meaning of “electrolyte solution”.
Page 2, Line 65:
Electrolyte solution of NaOH and hydrochloric acid (HCl) are introduced to remove the oxide layer on the surface of LMDs where NaOH solution was prepared by dissolving solid sodium hydroxide particles in deionized (DI) water and HCl solution was prepared by diluting concentrated hydrochloric acid with DI water.
2. Size dependency experiment, 117-126, Figure 2b:
The interpretation that the rotational friction against the sidewall is unsupported, repeating the experiment with different geometry test tubes would elucidate this.
Response 2:
We thank the reviewer for the comment. We have conducted additional experiments and added more discussions to explore this phenomenon, as given in Supporting Information S1. Also the following descriptions have been added to the main manuscript.
Page 4, Line 132:
When the volume of the EGaIn droplets is larger than 0.06 mL, the friction between the droplet and the sidewall may give a negative effect on the rotational speed of the droplet which was been elucidated by repeating the experiment with a bigger tube as discussed in Supporting Information S1.
Supporting Information S1: Self-rotation of EGaIn droplets in different diameter tubes
In order to investigate the mechanism of the self-rotational speed partially decreases as the droplet volume increases, we repeated the experiment using a larger diameter tube (diameter of 10 mm). As shown in Figure S1, at the beginning, the self-rotational speed of the droplet increases with the increase of the droplet volume, which is consistent with the performance in the 8 mm diameter tube. However, when the volume of EGaIn droplets is larger than 0.08 mL, the rotational speed decreases a little with the volume increases and eventually remains stable. In compared with 8 mm diameter tube, the 10 mm tube has a larger threshold 0.08 mL (the threshold for the 8 mm diameter tube is ~0.06 mL). These demonstrate that the diameter of the tube effects the self-rotational speed, that is to say, the friction between the droplets and the sidewall of the tube might have a negative influence on the self-rotational speed.
Figure S1. Plots of self-rotational speed vs. sizes of droplets.
3. Immersion dependency experiment, 126-136, figure 2b:
More evidence is required. If the Authors claim is true then the speed should maximize at or over 100% immersion as the ‘slip’ increases and until reducing with the increased viscous friction.
Response 3:
We thank the reviewer for the comment. We have conducted additional experiments and added the 100% immersed curve to figure 2b.
Figure 2. (b) Self-rotational speed vs. sizes of droplets plot, blue, red, yellow and purple curves indicate different depths of EGaIn droplets immersion in the NaOH solution, respectively.
4. Electrolyte concentration experiment, 135-143, Figure 2 c and d
More data points are required.
I do not believe the author’s interpretation explains the phenomena fully. There are insufficient Data points to properly determine response of the system.
a. I agree that the oxidation prevention is the likely the key mechanism increasing speed however, the authors use vague imprecise terminology.
i. I would assume the response to appear to be that of a step response (V = Vmax(1-e^(-x/c)), where x is the concentration and c a constant based on the friction co-efficient of the oxide and the equilibrium oxidation in the solution.
ii. It also seems that HCl has a minimum concentration.
Response 4:
We thank the reviewer for the comment. We have conducted additional experiments and revised the main manuscript according to the comments.
a. The following descriptions have been added to the main manuscript to provide a more distinct explanation of the effect of oxidation on self-rotational speed of EGaIn droplets.
Page 4, Line 152:
There are no self-rotation in our experiments was observed when we reduce the NaOH solution concentration to zero. That might be due to the fact that without NaOH, the oxide layer cannot be removed and the droplet becomes wrinkled [14, 17], eventually the friction between the droplet and the tube and the solution increases.
i. We have conducted additional experiments and more data have been supplemented in figure 2 c and d.
Figure 2. (c) Self-rotational speed vs. concentration of NaOH plot.
ii. We have conducted additional experiments and found that the concentration of hydrochloric acid (HCl) required to remove the oxide layer and enable the self-rotation of EGaIn droplet is at least 0.025 mol/L.
Figure 2. (d) Self-rotational speed vs. concentration of HCl plot.
5. 138-149: I believe you mean to say ‘NaOH water was substituted for DI water’ the way it is written currently is confusing to the reader. Although in context simply stating that NaOH concentration is reduced to zero is sufficient for the reader if the solution is previously defined (see earlier comment).
Response 5:
Thank you for your suggestion and the following descriptions have been fixed according to the comment and added to the main manuscript.
Page 4, Line 152:
There are no self-rotation in our experiments was observed when we reduce the NaOH solution concentration to zero. That might be due to the fact that without NaOH, the oxide layer cannot be removed and the droplet becomes wrinkled [14, 17], eventually the friction between the droplet and the tube and the solution increases.
6. Mixing experiment, Figure 4, 151-157: were different volumes, viscosities of mixing liquid?
a. How would this affect the speed of rotation?
Response 6:
We have fixed the main manuscript according to the comment and hope that the explanation clarifies the points we attempted to make.
Page 5, Line 184:
Based on the fact that LMDs are negligible soluble in most liquids, and have great surface tension and keep almost unmixed with the solvent [7], LMDs have great potential in the application of mixer. Therefore, we further investigated its application as a mixer based on the self-rotation phenomenon of LMDs in a rotating magnetic field, as shown in Figure 4a (also see Video S2). We applied the self-rotation of EGaIn droplets to mix two droplets of the same volume, similar viscosity, and different colors.
Page 6, Line 192:
The self-rotational speed of EGaIn droplet after mixing was ~70 RPM which is almost the same as that when the EGaIn droplet is not acting as a mixer shown in figure 2b.
7. As the authors mention cooling as one potential application the thermal response of the system should be tested.
Response 7:
Thank you for providing this interesting perspective and we have conducted an experiment to explore the application of self-rotation of liquid metal as a cooler. The following descriptions have been added to the main manuscript.
Page 1,Line 25:
We further demonstrate the ability of accelerating cooling and a mixer enabled by the self-rotation of a LMD.
Page 5, Line 170:
Cooling system is an import potential application of liquid metal, and here we demonstrate the ability of liquid metal self-rotation to accelerate liquid cooling. As shown in Figure 3a, we heated the 0.5 mol/L NaOH solution and 0.08 mL EGaIn in a tube with a heat gun (DH-HG2-2000, Delixi Electric). When the temperature of the solution reached about 40 ℃, a large number of bubbles were separated from the solution like boiling. We stopped heating and rotated the motor at 420 RPM and measured the temperature of the solution every 5 seconds. For comparison, the other tube was tested in the same way except that the permanent magnets in the device were removed, that is, the LMD did not self-rotate as the motor rotated. The temperature change is shown in figure 3b, the group in which the EGaIn droplet self-rotated cooled significantly faster than the group in which the EGaIn droplet did not rotate. After about 520 s, the group of EGaIn rotated cooled to room temperature (~24.2 ℃),and after another 130 s, the other group cooled to room temperature.
Figure 3. Cooler based on a self-rotating LMD. (a) Schematic illustration of the cooling setup. (b) Temperature of solution vs. time plot.
Page 6, Line 205:
Moreover, we demonstrated that such a technique can be used for the application of accelerating cooling and mixing liquids.

Reviewer 2 Report
This is an interesting paper, giving a sufficient description of the experiments performed and clear guidance of possible applications. However, the physical motivation and explanations of the driving mechanisms are not sufficient for a scientific publication. Similar experiments and theory have been published in the context of magnetic stirrers driven by rotating permanent magnets: e.g.,
- T. Beinerts, A. Bojarevičs, I. Bucenieks et al.: Use of Permanent Magnets in Electromagnetic Facilities for the Treatment of Aluminium Alloys, Metall and Materi Trans B, (2016), 47: 1626
- V. Dzelme, A. Jakovics, I. Bucenieks. Numerical modelling of liquid metal electromagnetic pump with rotating permanent magnets. doi:10.1088/1757-899X/424/1/012046
- and more.
Use of the terminology 'Self-rotation of Liquid Metal Droplets' is misleading, a simple 'rotation of liquid droplets by the action of permanent magnets' would be more appropriate.
The equation (1) is the integral form of the Faraday law, which does not explain the source of the rotating force. This force originates due to the phase shift of the imposed magnetic field and the induced current, otherwise there will be a zero value to the integral torque if taking the presented explanations only.
The crucial presence of the electrolytes in the experiment means the presence of chemical reactions. The droplet covered by oxides will rotate as a solid body if partially suspended in the surrounding water. The experiments with pure water presented by the authors seem to support this. The Figure 2b presents an experimental point at zero which is impossible - this is an extrapolation.
Overall, we would recommend this paper for publication after adding some clarifications about the physical nature of the observed phenomena.
Author Response
Response to Reviewer 2 Comments
1. Similar experiments and theory have been published in the context of magnetic stirrers driven by rotating permanent magnets: e.g.,
- T. Beinerts, A. Bojarevičs, I. Bucenieks et al.: Use of Permanent Magnets in Electromagnetic Facilities for the Treatment of Aluminium Alloys, Metall and Materi Trans B, (2016), 47: 1626
- V. Dzelme, A. Jakovics, I. Bucenieks. Numerical modelling of liquid metal electromagnetic pump with rotating permanent magnets. doi:10.1088/1757-899X/424/1/012046
- and more.
Response 1:
We thank the reviewer for the comment. The following references have been added to the main manuscript.
45. Beinerts, T.; Bojarevičs, A.; Bucenieks, I.; Gelfgat, Y.; Kaldre, I. Use of permanent magnets in electromagnetic facilities for the treatment of aluminum alloys. Metall. Mater. Trans. B 2016, 47(3), 1626-1633, DOI: 10.1007/s11663-016-0646-5.
46. Dzelme V.; Jakovics A.; Bucenieks I. Numerical modelling of liquid metal electromagnetic pump with rotating permanent magnets. Mater. Sci. Eng. 2018, 424(1): 012046, DOI: 10.1088/1757-899X/424/1/012046.
2. Use of the terminology 'Self-rotation of Liquid Metal Droplets' is misleading, a simple 'rotation of liquid droplets by the action of permanent magnets' would be more appropriate.
Response 2:
We thank the reviewer for the comment. The title has been fixed according to the comment.
3. The equation (1) is the integral form of the Faraday law, which does not explain the source of the rotating force. This force originates due to the phase shift of the imposed magnetic field and the induced current, otherwise there will be a zero value to the integral torque if taking the presented explanations only.
Response 3:
We thank the reviewer for the comment. The “phase shift” mentioned by the reviewer may be similar to the “relative motion” mentioned in the main manuscript. We have modified the main manuscript to misunderstanding. The following description have been added to the main manuscript to clarify the formation of torque.
Page 2, Line 86:
When the external magnetic field starts to rotate, relative motion (in other words, phase shift) is formed between the magnetic field and the EGaIn droplet, which further induces the change of the magnetic flux through the equivalent coils φ and eddy current I within the equivalent coils.
Page 3, Line 100:
According to Ampere’s law, the torque Me can be expressed as
(2)
where R is the radius of the equivalent coil, l is the length of the equivalent coil, respectively [42-46].
4. The crucial presence of the electrolytes in the experiment means the presence of chemical reactions. The droplet covered by oxides will rotate as a solid body if partially suspended in the surrounding water. The experiments with pure water presented by the authors seem to support this. The Figure 2b presents an experimental point at zero which is impossible - this is an extrapolation.
Response 4:
We thank the reviewer for the comment. The following have been revised to provide a more accurate description. Also figure 2b has been revised according to the comment.
Page 2, Line 53:
Therefore, we have been stimulated to explore novel methods for inducing self-rotation of LMDs, which are smooth, simple and steady, especially without introducing undesired violent chemical reaction. According to our previous report, we introduced a simple and violent chemical reaction-free method in which we utilized Lorenz force induced by magnetic fields to control the locomotion of LMDs [38].
Page 6, Line 200:
We have demonstrated a novel method to manipulate the self-rotation of LMDs by solely utilizing magnetic fields which is smooth, simple and steady, especially without the introduction of violent chemical reaction.
Figure 2. (b) Self-rotational speed vs. sizes of droplets plot, blue, red, yellow and purple curves indicate different depths of EGaIn droplets immersion in the NaOH solution, respectively.

Reviewer 3 Report
Title: Self-rotation of Liquid Metal Droplets Solely Driven by Magnetic Fields
Authors: Jian Shu et. al.
Authors in this article explained the theoretical conception regarding rotation of LMDs and provided limited validation. However, I do not find any significance in this work as authors already mentioned the finding of this article in their previously published article ref. 33, “Unconventional locomotion of liquid metal droplets driven by magnetic fields”. Only difference in this article is that the entire article is centered on rotation as oppose to locomotion but do not provide any rationale regarding existing current problems that they have solved in the field of rotation of LMDs. Therefore, I would like to advise editor that please reject and do not consider this article for publication unless authors provide major revision because in absence of major revision that clearly emphasizes on significance of this work, this article would be close repeat of their previous article if not duplication.
In addition, following comments needs to address in major revision.
1. Introduction does say that liquid metal droplets and their rotation are useful but authors did not mention along with citation why it is important to induce self-rotation of LMDs which are smooth, simple and steady, especially without introducing undesired chemical reaction. I think the introduction in not justifying why it is important to do work that author did. Adding this will be necessary for guiding the readers of this journal article.
2. How did the speed of magnetic rotation and droplet rotation measure?
3. Please clarify why it is obvious that using NaOH solution with a high concentration can leads to a faster rotational 136 speed.
4. Please add citation for “NaOH remove the oxide layer on the surface of EGaIn droplets”
5. Author previously published the same observation in their reference 33 and here is the quote from their previous paper. “Interestingly, we also observed the self-rotation of the EGaIn droplet while it is travelling along the PMMA channel, as shown in Fig. 1G, in which we used red particles to coat the surface of the EGaIn droplet to clearly show the rotation (also see Movie S3, ESI†). We believe that the self-rotation can be attributed to the fact that the magnetic flux density experienced by the two hemispheres of the EGaIn droplet is uneven due to the imperfect alignment when a magnet passed under the EGaIn droplet, as shown in the cross-sectional schematic given in Fig. 1H. Such a misalignment will induce a larger Lorenz force on one side of the hemisphere and generate a torque in the horizontal plane to rotate the droplet. The mechanism for explaining the self-rotation phenomenon was further validated by conducting experiments using channels with different configurations to induce different asymmetry scenarios of magnetic fields on the EGaIn droplet, as discussed in the ESI S4.†”. Please clarify that what is new finding reported in this article?
6. Clarify this sentence “Further study has demonstrated that LMDs coating 43 advisable nanoparticles can be actuated by bubbles generated through an electrochemically reaction 44 [28-30].”
7. Change analyze to analysis in line 106.
8. In addition, significant portion of text in this article matches with the previous article (as it is) and does not justify why the rotation of droplet using the way they have done will be useful of beneficial.
9. In this paper, I do not find any additional finding or in-depth analysis which will add to the understanding which authors already published.
Author Response
Response to Reviewer 2 Comments
1. Introduction does say that liquid metal droplets and their rotation are useful but authors did not mention along with citation why it is important to induce self-rotation of LMDs which are smooth, simple and steady, especially without introducing undesired chemical reaction. I think the introduction in not justifying why it is important to do work that author did. Adding this will be necessary for guiding the readers of this journal article.
Response 1:
We thank the reviewer for the comment. The following descriptions have been added to the main manuscript to illustrate the importance of inducing self-rotation of LMDs which are smooth, simple and steady, especially without introducing undesired chemical reaction.
Page 2, Line 46:
Chaotic advection is the key mechanism for enabling applications such as heat transfer, fluid mixing and chemical reaction improvement [24, 31-35], in particular, in some areas including microfluidic systems, chemical and biological transport et. al., the smooth and steady methods for inducing chaotic advection is needed and inexpensive and simple systems are urgently required [36-39]. Self-rotation of LMDs with negligible solubility in most solvents, may be a promising candidate for solution [7]. Nevertheless, to the best knowledge of the authors, there are still lack of investigations focused on the self-rotational motion of LMDs.
2. How did the speed of magnetic rotation and droplet rotation measure?
Response 2:
We thank the reviewer for the comment. The following descriptions have been added to the main manuscript to introduce the measurement of rotational speed.
Page 2, Line 68:
A small amount of fine phosphors was sprinkled on the LMDs to facilitate the measurement of their rotational speed.
Page 2, Line 75:
A pair of permanent magnets were fixed to the aluminum frame, and the aluminum frame was connected to the output shaft of a DC motor (Leadshine 57HS09) whose speed and direction are controlled by an MCU (Arduino Carduino UNO R3).
3. Please clarify why it is obvious that using NaOH solution with a high concentration can leads to a faster rotational speed.
Response 3:
We thank the reviewer for the comment. The following descriptions have been added to the main manuscript to clarify the influence of the NaOH solution concentration on the LMD self-rotational speed.
Page 4, Line 149:
We found that using NaOH solution with a high concentration can leads to a faster rotational speed until the concentration of NaOH solution reaches 0.03 mol/L. When the concentration of NaOH solution exceeds 0.03 mol/L, the rotational speed of droplets no longer increases with the increase of concentration and remains almost a constant. There are no self-rotation in our experiments was observed when we reduce the NaOH solution concentration to zero. That might be due to the fact that without NaOH, the oxide layer cannot be removed and the droplet becomes wrinkled [14, 17], eventually the friction between the droplet and the tube and the solution increases. With the increase of the concentration of NaOH solution, the oxide layer gradually removed; the friction gradually decreased; and the rotational also increases. However, when the concentration reaches the threshold (0.03 mol/L), the oxide layer is almost completely removed, and the speed no longer increases with the concentration increase.
Figure 2. (c) Self-rotational speed vs. concentration of NaOH plot.
4. Please add citation for “NaOH remove the oxide layer on the surface of EGaIn droplets”.
Response 4:
We thank the reviewer for the comment. Citations have been added to the main manuscript. Please see point 3 above.
5. Author previously published the same observation in their reference 33 and here is the quote from their previous paper. “Interestingly, we also observed the self-rotation of the EGaIn droplet while it is travelling along the PMMA channel, as shown in Fig. 1G, in which we used red particles to coat the surface of the EGaIn droplet to clearly show the rotation (also see Movie S3, ESI†). We believe that the self-rotation can be attributed to the fact that the magnetic flux density experienced by the two hemispheres of the EGaIn droplet is uneven due to the imperfect alignment when a magnet passed under the EGaIn droplet, as shown in the cross-sectional schematic given in Fig. 1H. Such a misalignment will induce a larger Lorenz force on one side of the hemisphere and generate a torque in the horizontal plane to rotate the droplet. The mechanism for explaining the self-rotation phenomenon was further validated by conducting experiments using channels with different configurations to induce different asymmetry scenarios of magnetic fields on the EGaIn droplet, as discussed in the ESI S4.†”. Please clarify that what is new finding reported in this article?
Response 5:
We thank the reviewer for the comment. There is no denying that the self-rotation of the liquid metal droplets (LMDs) driven by magnetic fields is inspired by the unconventional locomotion of liquid droplets in magnetic fields. However, the use of a coaxial rotating vertical magnetic field to actuate the self-rotation of LMDs is low in efficient and controllability. In view of the application prospects of the LMDs self-rotation in liquid mixing, chip cooling, lab-on-a-chip and soft robotics, we proposed a more efficient, controllable and convenient self-rotation actuation method. On this basis, we tested the ability of LMDs self-rotation to accelerate heat dissipation and its application as a mixer.
6. Clarify this sentence “Further study has demonstrated that LMDs coating advisable nanoparticles can be actuated by bubbles generated through an electrochemically reaction [28-30].”
Response 6:
We thank the reviewer for the comment. Tang et al. found that liquid metal droplets encasing in WO3 nanoparticles placed in H2O2 solution can be propelled with the illumination of UV light. The driving force comes from the oxygen bubbles which is generated by the photochemical reaction triggered by the semiconducting WO3 (Tang, Appl. Phys. Lett. 2013). We have modified the original sentence and added it to the main manuscript.
Page 1, Line 43:
Further studies have demonstrated that LMDs coating modest nanoparticles can be propelled by bubbles generated through photochemically reactions [28-30].
7. Change analyze to analysis in line 106.
Response 7:
We thank the reviewer for the comment and the typos in the main manuscript have been fixed according to the comment.
8. In addition, significant portion of text in this article matches with the previous article (as it is) and does not justify why the rotation of droplet using the way they have done will be useful of beneficial.
Response 8:
We thank the reviewer for the comment. We have demonstrated a novel method to manipulate the self-rotation of LMDs by solely utilizing magnetic fields which is smooth, simple and without the introduction of violent chemical reactions, especially, it is a non-contact driving method. We believe that it has the potential to be used not only in macroscopic liquid mixing and chip cooling in closed environment, but also can make progress in the frontier fields of lab-on-a-chip and soft robotics.
9. In this paper, I do not find any additional finding or in-depth analysis which will add to the understanding which authors already published.
Response 9:
We thank the reviewer for the comment. As mentioned by the reviewer, in the previous study we found some unconventional locomotion of LMDs in the rotating magnetic fields, and proposed a new LMD actuating method, which is similar to the method used in this paper. However, in view of the current situation that it is difficult to directly obtain applications of rotating magnetic field actuating LMDs locomotion, we found that magnetic driven self-rotation seems to be easily transplanted, controlled and used by other systems. We have presented the application as a mixer to prove this statement. To emphasize our point, the following description have been added to the main manuscript to demonstrate the application of LMDs self-rotation in accelerating liquid cooling.
Page 5, Line 170:
Cooling system is an import potential application of liquid metal, and here we demonstrate the ability of liquid metal self-rotation to accelerate liquid cooling. As shown in Figure 3a, we heated the 0.5 mol/L NaOH solution and 0.08 mL EGaIn in a tube with a heat gun (DH-HG2-2000, Delixi Electric). When the temperature of the solution reached about 40 ℃, a large number of bubbles were separated from the solution like boiling. We stopped heating and rotated the motor at 420 RPM and measured the temperature of the solution every 5 seconds. For comparison, the other tube was tested in the same way except that the permanent magnets in the device were removed, that is, the LMD did not self-rotate as the motor rotated. The temperature change is shown in figure 3b, the group in which the EGaIn droplet self-rotated cooled significantly faster than the group in which the EGaIn droplet did not rotate. After about 520 s, the group of EGaIn rotated cooled to room temperature (~24.2 ℃),and after another 130 s, the other group cooled to room temperature.
Figure 3. Cooler based on a self-rotating LMD. (a) Schematic illustration of the cooling setup. (b) Temperature of solution vs. time plot.

Round 2
Reviewer 1 Report
The authors have adressed all issues that I observed regarding the manuscript.
However, I do have one remaining concern regarding data.
In Figure 2b, for the 100% coverage depth, the established trends seems to reverse (maximal RPM was recorded at 0.04 mol/L), the authors do not discuss this.
Perhaps more datapoints are required to understand the mechanism?
or Perhaps modeling the frictions as a function of depth droplet size would reveal the answer.
Aside from this remaining quandary I beleive the manuscript is warranted for publication.
Author Response
Response to Reviewer 1 Comments
1. The authors have addressed all issues that I observed regarding the manuscript.
However, I do have one remaining concern regarding data. In Figure 2b, for the 100% coverage depth, the established trends seems to reverse (maximal RPM was recorded at 0.04 mol/L), the authors do not discuss this. Perhaps more datapoints are required to understand the mechanism? Or perhaps modeling the frictions as a function of depth droplet size would reveal the answer.
Aside from this remaining quandary I believe the manuscript is warranted for publication.
Response 1:
We thank the reviewer for the comment. The following descriptions have been added to the main manuscript to discuss the unconventional trend ant 100% coverage depth.
Page 4, Line 150:
Interestingly, we observed that when the droplet was 100% immersed, the droplet reached its maximum rotational speed when the volume was increased to 0.04 mL, which is smaller than that of other immersion depths (0.06 mL, 0.06 mL, 0.08 mL). We believe this is due to the fact that the significant increase in rotational speed at 100% immersion flattens the droplets and therefore, increases the friction between the droplets and the tube.

Reviewer 2 Report
The responses to the reviewer comments are sufficient to recommend the paper for publication in this journal.
Author Response
We thank the reviewer for the comments.
Reviewer 3 Report
Dear Editor,
The revision has significantly improved but I still need following minor changes necessary before acceptance.
In Introduction, the reasons why authors method is required are not listed. They said why LMD rotation and its application but did not highlight current method of doing LMD and how or what problem their technique is going to solve/overcome.
It is highly unsuitable to say that "A small amount of fine phosphors was sprinkled on the LMDs to facilitate the measurement of their rotational speed." Please provide brief description so that readers could be able to reproduce of apply similar techniques to other research.
I think, minor revision fulfilling these two comments will make the article suitable and readable to audience of your journal.
Author Response
Response to Reviewer 3 Comments
1. In Introduction, the reasons why authors’ method is required are not listed. They said why LMD rotation and its application but did not highlight current method of doing LMD and how or what problem their technique is going to solve/overcome.
Response 1:
We thank the reviewer for the comment. The following descriptions have been added to the main manuscript to illustrate the problem we are going to solve.
Page 2, Line 53:
Self-rotation of LMDs has the potential to be widely used in fluid cooling and mixing. Therefore, we have been motivated to explore novel methods for inducing self-rotation of LMDs that are smooth, simple, steady, and especially without introducing undesired violent chemical reaction.
Page 2, Line 63:
Moreover, we demonstrated applications of accelerating cooling and mixing liquids based on the self-rotational LMDs.
2. It is highly unsuitable to say that "A small amount of fine phosphors was sprinkled on the LMDs to facilitate the measurement of their rotational speed." Please provide brief description so that readers could be able to reproduce of apply similar techniques to other research.
Response 2:
We thank the reviewer for the comment and the following descriptions have been revised to clarify the experimental procedure.
Page 2, Line 70:
A small amount (~2 mg) of fine phosphors (Juen technology Co. Ltd, China) was sprinkled on the LMDs so that we can find some easily identifiable points for calculating the rotational speed of LMDs.
